# Anomaly Detection Using Electric Impedance Tomography Based on Real and Imaginary Images

**DOI:** 10.3390/s20071907

**Published:** 2020-03-30

**Authors:** Imam Sapuan, Moh Yasin, Khusnul Ain, Retna Apsari

**Affiliations:** 1Department of Physic, Faculty of Science and Technology, Universitas Airlangga, Surabaya 60115, Indonesia; i_sapuan@fst.unair.ac.id (I.S.); yasin@fst.unair.ac.id (M.Y.);; 2Biomedical Engineering, Faculty of Sciences and Technology, Universitas Airlangga, Surabaya 60115, Indonesia

**Keywords:** Electrical Impedance Tomography, detection of anomaly, resistive and capacitive image, real and imaginary image

## Abstract

This research offers a method for separating the components of tissue impedance, namely resistance and capacitive reactance. Two objects that have similar impedance or low contrast can be improved through separating the real and imaginary images. This method requires an Electrical Impedance Tomography (EIT) device. EIT can obtain potential data and the phase angle between the current and the potential measured. In the future, the device is very suitable for imaging organs in the thorax and abdomen that have the same impedance but different resistance and capacitive reactance. This device consists of programmable generators, Voltage Controlled Current Source (VCCS), mulptiplexer-demultiplexer potential meters, and phase meters. Data collecting was done by employing neighboring, while reconstruction was used the linear back-projection method from two different data frequencies, namely 10 kHz and 100 kHz. Phantom used in this experiment consists of distillated water and a carrot as an anomaly. Potential and phase data from the device is reconstructed to produce impedance, real, and imaginary images. Image analysis is performed by comparing the three images to the phantom. The experimental results show that the device is reliable.

## 1. Introduction

Mapping the electrical impedance distribution of an object is the basic principle of Electrical Impedance Tomography (EIT) technology. The impedance of human body tissue is able to provide information about the physiological and pathological properties of the tissue [1,2,3]. Both of these properties are related to the information of medical applications [1,2,4,5,6,7,8,9,10,11,12]. This technology is carried out by injecting electrical currents and measuring the voltage generated through the electrodes. The measurement of voltage with forward-problem results in voltage distribution. An inverse problem is the reconstruction process to obtain an impedance distribution [3,4,13,14,15,16]. The body tissue impedance consists of two parts, which are real resistance and imaginary capacitive reactance [3,14,16,17]. The present study separates the distribution of those two impedance components.

The EIT imaging technique is noninvasive, portable, simple, fast, easy to use, economical, comfortable for the patient’s body, it also does not cause ionizing radiation [18,19]. This technology has been applied in many clinical applications such as detection for breast cancer [6], nerve activity in the brain [7], respiratory disorders [10,20], and lung function detection [4]. 

The basic principle of EIT in detecting abnormalities is based on the impedance differences between normal tissue and anomalies. The EIT is able to detect abnormalities in body tissue if both impedances are different [4,5,6,7]. For the benefit of EIT researches, normal tissue objects and anomalies use phantoms that are made of materials with different impedances. Some of EIT applications include detection of breast cancer, normal breast tissue, and cancer using different impedances [3,14,21]. 

Multi-frequency Electrical Impedance Tomography (Mf-EIT) [3,14,16] is a type of EIT which produces many structural images for different frequencies. The resulting image is an impedance distribution as a function of frequency. The images are impedance images from 5, 10, 25, 50, 75, and 100 kHz [16]. These results show the image of resistivity inhomogeneity is caused by differences. 

The weakness of the impedance image generated from the MF-EIT system [8,13] is its inability to distinguish objects that have the same impedance but are different tissues. Some tissues that have the similar impedance include liver and lungs with the impedance value of 1.622 Ω and 1.624 Ω, heart and deflated lung with the impedance value of 1.001 Ω and 1.077 Ω at the frequency of 10 khz. While the impedance value of colon and cervix are respectively 3.3168 Ω and 3.2875 Ω at the frequency of 100 kHz. The data was obtained from a simulation developed by C. Gabriel [22]. The impedance of these organs has different conductivity and permittivity even with the similar impedance. As a result, the Mf-EIT system does not show the differences in these kind of tissue. However, if real and imaginary images are produced, it will be possible to show the difference between the two objects which have the same impedance.

To overcome the disadvantage of the Mf-EIT system, we have built EIT which can break down two components of the body tissue impedance, namely resistive resistance and capacitance [23,24,25]. The concept used to break down two impedance components is by using the resistor capasitor RC series model [23,25], which has been applied to Bioelectrical Impedance Analysis (BIA) devices. This method uses the properties of the circuit resistance, capacitance, impedance, and phase shift in the alternating current AC circuit. The whole series and or parallel circuits consisting of resistors R and capacitors C has an equivalency to RC series [23,26]. The circuit consists of RC components in series and or parallel relationships, regardless of the number and value of RC it could be simplified in the final form of RC series with a certain value. Body tissue composed of unknown R and C, which is analogous to the concept, could be replaced with RC series with the equivalent R and C values.

The practice of this method begins by assuming the body tissue as a RC series. Furthermore, the body tissue is injected with AC current. At some point voltage and "phase difference" will be measured. The study used the Neighboring data collection method with 16 electrodes. This method works by injecting current and measuring the voltage between the two closest electrodes [4]. The intended voltage is the difference in electrical potential at two specific points when current is injected into the object. Meanwhile, the phase difference is the phase difference between the voltage and the injection electric current. From these two data, impedance, resistive and capacitive reactance can be calculated. This concept is applied to build EIT in mapping the impedance, resistive and capacitive reactance of body tissues. In the reconstruction process, the resistive reactance potential produces a real image, capacitive reactance produces a capacitive image, and the impedance produces an impedance image.

The study has designed and developed the EIT device without involving a high-speed data acquisition system but it can produce impedance image components that include real and imaginary images. To show anomalies in normal tissues, EIT utilizes a functional image [27]. The image uses two different frequencies of injection current [4,27]. The research used current sources at the frequency 10 kHz and 100 kHz. EIT produces three functional images, namely impedance image, resistive image, and capacitive image. These three images can be used in the processes of identifying and analyzing tissues. Moreover, with these three images, tissue diagnostics and analysis are improved to be more accurate than by using only one impedance image.

## 2. Materials and Methods

Electrical Impedance Tomography (EIT) focuses on the separation of electrical impedance components, which are the resistance (part of the real impedance) and the capacitive reactance (part of the imaginary impedance) [3]. The experimental EIT flowchart in this study was shown in Figure 1. The study began with the design of the EIT data acquisition system. Next, we examined the performance of the equipment tested in each section, and carried out integration and synchronization. Synchronization and integration combine all equipment including hardware and software into one EIT system. The last stage was an experiment by testing the performance of the EIT system to obtain real and imaginary images.

The real and imaginary part of impedance can be measured using Fourier transforms. The method is simple, however we try to avoid data acquisition systems or high-speed sampling systems in this study. The method for obtaining resistance and reactance of an object is to inject AC into the object. We then measure the voltage and the phase difference between the current injection and the voltage. These steps are performed for all electrodes attached to the object. This phase difference measurement uses the AC series circuit concept [23]. Various combined RC series and parallel circuits can be replaced with a value equivalent to a single RC series value [23,26]. Equations (1) and (2) express the relationship between impedance, phase difference, resistance, and capacitive reactance in an RC series circuit.
(1)$$Z=\sqrt{R^{2}+X_{c}^{2}}$$
(2)$$\tan\;=\frac{X_{c}}{R}$$
where *Z* is the impedance, R is the resistance, X_C_ is capacitive reactance, and θ is the phase difference. These equations can be used to determine the value of R and X_C_. The value of R and X_C_ are expressed by Equations (3) and (4). The relation of R, Xc, and Z is shown in Figure 2.
R = Z cos θ
(3)

X_c_ = Z sin θ
(4)

Figure 3 shows that resistance and capacitive reactance are obtained by measuring the voltage at a pair of adjacent electrodes, and the phase difference (θ) between the injection current (I) and the resulting voltage. The impedance between the two electrodes is assumed to be the RC series value. The result of measuring the potential between the two electrodes is the voltage (V). To get a current injection, a current to voltage converter circuit is required. This circuit is to convert current signals into voltage signals V_I_. The voltage signals V_I_ and V were connected to the phase detector input circuit. The output of the phase detector displayed the direct current DC voltage as a function of the phase difference between the two signals. The phase difference detected by the phase detector is the same as the output voltage.

The injection current measurement results (I) and voltage (V) are used to determine the magnitude of the impedance (Z). The phase difference (θ) is measured using a phase detector. Resistance (R) and capacitive reactance (Xc) could be calculated using Equations (3) and (4). The EIT data acquisition system is needed to apply the concept of resistance and capacitive reactance measurement. The block diagram of the data acquisition system was shown in Figure 4.

This system consisted of oscillator, buffer, and Voltage Controlled Current Source (VCCS), multiplexer of current injection, object (phantom), de-multiplexer of voltage measurement, current to voltage converter, differential amplifier, AD 8302 phase detector by ATI Technologies Inc (Markham, Canada), and Arduino Mega.

### 2.1. Voltage Controlled Current Source (VCCS) 

VCCS has a very important role as a constant current source. Figure 5 shows a constant current source generator scheme consisting of AD 9850 modules by Digi-Key Electronics (Thief River Falls, MN, USA), DC blocks, amplifier circuits, buffers, and VCCS double op-amps. The working principle of VCCS starts from the sinusoidal DC signal generated by the AD 9850 modules. Then the sinusoidal DC signal is passed through the DC block circuit to generate an AC signal. The non-inverting op-amp amplifies the DC block output signal as VCCS input. The VCCS input voltage is proportional to the current generated by VCCS. The signal amplifier circuit is used to increase the current generated by VCCS. This VCCS generates sinusoidal currents of 1 mA, 10 to 100 kHz. To keep the AC current constant, the buffer is placed before the VCCS double op-amp.

The maximum voltage generated by the AD 9850 module is 0.313 volts. The non-inverting op-amp amplifier functions to amplify the signal output of the 9850 AD module. The magnitude of this amplification will affect the amount of current generated by VCCS. The resistor in VCCS dual op-Amp Amp R_F_, Rs, and R_1_ have the same value of 2 kΩ. The amount of gain depends on the amount of current needed. The VCCS circuit is loaded R_L_ = 1, 2, and 3 kΩ. VCCS characterization is related to the stability of the current generated with respect to frequency and load.

### 2.2. Electric Mechanism of Current Injection and Voltage Measurement, Current to Voltage Circuit to Measure Phase Difference

This study used 16 electrodes namely electrodes E_1_, E_2_, E_3,_…, E_16_. The process began with injecting an electric current through the adjacent electrode E_1_ signal - E_2_ ground. At the same time, the voltage in the pair of adjacent electrodes (E_2_–E_3,_ E_3_–E_4,_ E_4_–E_5_,…_,_ E_16_–E_1_) were measured. Next, the current was injected into the E_2_ signal - E_3_ ground electrode and the voltage in the pair of adjacent electrodes (E_3_–E_4,_ E_4_–E_5,_ E_5_–E_6_, …_,_ E_1_–E_2_) were again measured. This process was repeated until all adjacent electrodes became current injection electrodes.

#### 2.2.1. Electronic mechanism for Current Injection

The process of EIT current injection was carried out by an electronic mechanism. This electronic mechanism is shown in Figure 6. (current part). The selection of the current injector electrode on the object used one IC4052 de-multiplexer and two IC 4051 multiplexers. The electrode mechanism for measuring potential differences between two adjacent electrodes was controlled by two IC 4051 de-multiplexer. The designed electric mechanism scheme is shown in Figure 6.

For technical reasons, the process of current injection and voltage measurement in this system was divided into two mechanisms. In the first mechanism, odd electrodes were used as the signal while even electrodes used as ground. On the other hand, in second mechanism, odd electrodes were used as ground and even electrodes used as signal. 

The first mechanism process was started by injecting electric current to the adjacent electrodes which are E_1_ signal–E_2_ ground, followed by measuring the voltage of the adjacent electrodes pair (E_2_–E_3,_ E_3_–E_4,_ E_4_–E_5_, …_,_ E_16_–E_1_). The next processes were injecting current to E_3_ signal–E_4_ ground and measuring the voltage of the adjacent electrode (E_4_–E_5,_ E_5_–E_6,_ E_6_–E_7_,…_,_ E_2_–E_3_). This process was repeated until all adjacent electrodes became current injection electrodes.

In Figure 6, the process of injecting electric currents was started from the in-signal connected to 4052 through x_1_ and the signal exited through via Z_1_. The relationship between x_1_ and Z_1_ was controlled by an electrical mechanism by Control Select CS, namely INH = 0, B_0_ = 0, and A_0_ = 1. Furthermore, Z_1_ was connected to input Y on IC I 4051, through outputs (X_1_, X_2,_ X_3_, X_4,_ X_5_, X_6,_ X_7_, X_8_) which then were connected to electrodes for current injection to objects (E1, E3, E5, E7, E9, E11, E13, E15) and then paired with electrodes (E2, E4, E6, E8, E10, E12, E14, E16) which were forwarded to ground. The connection with ground through de-multiplexer IC II 4051 (X_1_, X_2,_ X_3_, X_4,_ X_5_, X_6,_ X_7_, X_8_) is the input while Y is the output. The set up of the electrical mechanism on IC I 4051 was done through CS (A_1_, B_1_ C_1_) while IC II 4051 was done through CS (A_2_, B_2_ C_2_). Y IC II 4051 pin was connected with Z_2_ IC 4052 as input that was connected to output y_1_. Finally, the current through the resistor was connected to ground. The resistor which was passed by the current injection was then connected in parallel with the op-amp amplifier, so that at the output of the op-amp the value and the shape of the injection current were obtained. The op-amp amplifier circuit in this position functions as a current to voltage (C to V) circuit. 

The second mechanism process was started by injecting electric current to the adjacent electrodes which are E_2_ signal–E_3_ ground, then measuring the voltage of the adjacent electrodes pair (E_3_–E_4,_ E_4_–E_5_, E_5_–E_6_…_,_ E_1_–E_2_). The process of current injection was continued on the E_4_ signal–E_5_ ground, and followed by measuring the voltage on the pair of adjacent electrodes which were (E_5_–E_6,_ E_6_–E_7,_ E_7_–E_8_, …_,_ E_1_–E_2_). This process was repeated until all adjacent electrodes became current injection electrodes.

In Figure 6, the process of injecting electric currents started from in-signal that was connected to 4052 through y_0_ and exited through Z_2_. The relationship between y_0_ with Z_2_ was controlled by electrical mechanism by CS, namely INH = 0, B_0_ = 0, and A_0_ = 0. Furthermore, Z_2_ was connected to the Y input on IC II 4051, through the outputs (X_1_, X_2,_ X_3_, X_4,_ X_5_, X_6,_ X_7_, X_8_) which were connected to the electrodes for the current injection to the objects alternately (E_2_, E_4_, E_6_, E_8,_ E_10_, E_12,_ E_14_, E_16_) and paired with electrodes (E_1_, E_3_, E_5_, E_7,_ E_9_, E_11,_ E_13_, E_15_) which then were forwarded to ground. The connection with ground via IC I 4051 (X_1_, X_2,_ X_3_, X_4,_ X_5_, X_6,_ X_7_, X_8_) as inputs and Y as output. The setting of the electrical mechanism on IC I 4051 was done through CS (A_1_, B_1_ C_1_) and on IC II 4051 was done through CS (A_2_, B_2_ C_2_). Next, Y IC I 4051 pin was connected to Z1 IC 4052 as input connected to output x_0_, then the current through the resistor was connected to ground. Similar to the first mechanism, the resistor which was passed by the current injection current was then connected in parallel with the op-amp amplifier, so that on the output the value and form of the current injection were obtained. The op-amp amplifier circuit in this position functions as a current to voltage (C to V) circuits. This series of C to V generated value and form of current signals.

#### 2.2.2. Electronic Mechanism of Voltage Measurement

The voltage measurement mechanism of the EIT system was electrically controlled. The electric circuit scheme of the voltage measurement process is shown in Figure 6. (voltage part). The circuit used was an AD 8039 op-amp buffer and two IC 4051 multiplexers. All electrodes to measure the voltage were connected to the buffer circuit first before being connected to a multiplexer. Odd electrodes (E_1_, E_3_, E_5_, E_7,_ E_9_, E_11,_ E_13_, E_15_) were each connected to a buffer then to the IC III 4051 multiplexer, respectively, at the inputs (X_1_, X_2,_ X_3_, X_4,_ X_5_, X_6,_ X_7_, X_8_). Next, the even electrodes (E_2_, E_4_, E_6_, E_8,_ E_10_, E_12,_ E_14_, E_16_) were also connected with a buffer then to the IC IV 4051 multiplexer, respectively, at the inputs (X_1_, X_2,_ X_3_, X_4,_ X_5_, X_6,_ X_7_, X_8_). 

Voltage measurement on the electrode pair was carried out after injecting the current. The selection of the active odd electrode was controlled by IC III 4051 through CS (A_3_, B_3_, C_3_) while the selection of the active even electrode was controlled by IC IV 4051 through CS (A_4_, B_4_, C_4_). In order to measure the voltage between E1–E2, electrode 1 and electrode 2 were activated by giving CS values on IC III 4051 (A_3_ = 0, B_3_ = 0, C_3_ = 0) and IC IV 4051 (A_4_ = 0, B_4_ = 0, C_4_ = 0). Same methods were done to activate other electrodes and measure the voltage, which was to change both CS inputs simultaneously. 

Output pin Y IC III 4051 and Y IC IV 4051 were connected with differentiator amplifiers which produced voltage value and form between the two adjacent electrodes. The value and form of this signal is called the voltage signal.

#### 2.2.3. Measurement of Voltage and Phase Difference 

Figure 7 shows a circuit schematic for the measurement of voltage and phase difference. The output voltage signal in the scheme is a sinusoidal signal differential between two adjacent electrodes. These two electrodes were used to measure the potential difference in an object caused by an injection current. In the application, the output voltage signals were connected to two connections. First, it was connected to the root mean square RMS to DC input. This RMS to DC circuit is also called an RMS to DC converter circuit. The input signal of this circuit is sinusoidal, so the output voltage is a DC voltage of V_rms_ from the potential difference measured on the object.

Second, the output voltage signal was connected to input A in AD8302 phase detector. This connection functions as a comparison of current signals output which was connected to input B in the AD8302 phase detector. Current signals and voltage signals provide information of phase difference between current injection and voltage. The phase detector output is a DC voltage, the value of which is proportional to the phase difference of the two signals. The phase difference of the two signals which was detected by the AD8302 phase detector is expressed by Equation (5). The value of this phase corresponds to the scheme and data-sheet components of AD8302. In the data sheet, *V_PSET_* is connected to the voltage reference in the module. The equation is to generate an AD 8302 module output voltage, as a measured phase.
(5)$$\left|\frac{900mV-V_{PSET}\left(V\right)}{10mV{/}^{{}^{\circ}}}+90^{{}^{\circ}}\right|=Phase\;sp{(}^{{}^{\circ}})$$

The circuit scheme (see Figure 7) has two functions that have to meet certain criteria. To measure the voltage of RMS to DC circuit, it has to meet the voltage stability function at the required frequency. To achieve this goal, the experiment was done to observe the stability of the output and input voltage at a certain value of the RMS to DC circuit at various frequencies. Then the experimental data were presented by a graph of the relationship between input and output voltages at various frequencies. The tools used in this performance test include a signal generator, RMS to DC circuit and digital storage oscilloscope BK Precision 2542 B.

The second function is to measure the phase difference of two sinusoidal input signals. The second parameter that needs to be known is the linearity of the phase difference conversion to the AD8302 phase detector output voltage. The tools used to determine the characteristics of this component are MyDAQ module, AD8302 phase detector, digital storage oscilloscope BK Precision 2542 B, and voltmeters. This experiment used the MyDAQ signal generator to produce two sinusoidal signals of phase difference. The experiment process was started with generating signals 1 and 2 at the same frequency, while in signal 2, the various phase difference φ was added. At signal 1, V(t) = A sin(ωt) serves as A input and at signal 2 V(t) = A sin(ωt + φ) served as B input in the phase detector. The AD8302 phase detector output is a DC voltage that can be measured with a multi-meter. The phase detector output voltage is a linear function in phase difference of the two signals. The setup of the Phase Detector linearity test tool is shown in Figure 8. 

### 2.3. Reconstruction

The image reconstruction was done by using the reconstruction method of Linear Back Projection (LBP). The algorithm of this method is based on a collection of neighboring data with circular geometry and equipotential based. The equipotential-based Linear Back Projection algorithm was initially proposed by Barber-Brown [4] which in matrix notation can be expressed in Equation (6).
(6)$${\left[{\delta \rho }_{n}\right]}_{\left(\mathrm{px}1\right)}={\left[F\right]}_{\left(\mathrm{pxp}\right)}{\left[B\right]}_{\left({\mathrm{pxq}}^{2}\right)}{\left[{\delta V}_{n}\right]}_{\left(q^{2}x1\right)}$$
where p is the number of elements and q is the number of electrodes, [B] is the back projection weight matrix, [F] is the representative matrix of filters, $\left[{\delta V}_{n}\right]$ is the potential change in normalized boundary, and $\left[{\delta \rho }_{n}\right]$ is a normalized change in impedance distribution.

The linearization method assumes that potential boundary changes are a linear function of changes in conductivity [4], so Equation (6) can be written in matrix as stated:(7)$${\left[\delta V\right]}_{\left(q^{2}x1\right)}={\left[S\right]}_{\left(q^{2}\mathrm{xp}\right)}{\left[\delta \rho \right]}_{\left(\mathrm{pxp}\right)}$$
where [δV] is the potential boundary change, [S] is the sensitivity matrix, and [δρ] is the resistivity change. In order to obtain the sensitivity matrix [*S*], a variation of δρ on all ρ in forward problem model. Equation (7) can be solved after the matrix [S] was measured by algebraic manipulation. However, because the matrix [S] is not square, [*δ*ρ] cannot be obtained directly.
(8)$${\left[S\right]}^{T}{}_{\left({\mathrm{pxq}}^{2}\right)}{\left[\delta V\right]}_{\left(q^{2}x1\right)}={\left[S\right]}^{T}{}_{\left({\mathrm{pxq}}^{2}\right)}{\left[S\right]}_{\left(q^{2}\mathrm{xp}\right)}{\left[\delta \rho \right]}_{\left(\mathrm{pxp}\right)}$$
(9)$${\left[\delta V\right]}_{\left(q^{2}x1\right)}={\left({\left[S\right]}^{T}\left[S\right]\right)}^{-1}{}_{\left(\mathrm{pxp}\right)}{\left[S\right]}^{T}{}_{\left({\mathrm{pxq}}^{2}\right)}{\left[\delta \rho \right]}_{\left(\mathrm{pxp}\right)}$$

Generally, ${\left[S\right]}^{T}\left[S\right]$ is a singular matrix so that the matrix does not have inverse. To solve these problems, Tikhonov regularization can be used so that the matrix has inverse.
(10)$${\left[\delta p\right]}_{\left(q^{2}x1\right)}={\left({\left[S\right]}^{T}\left[S\right]+\alpha I\right)}^{-1}{}_{\left(\mathrm{pxp}\right)}{\left[S\right]}^{T}{}_{\left({\mathrm{pxq}}^{2}\right)}{\left[\delta V\right]}_{\left(\mathrm{pxp}\right)}$$
where α is the regulatory parameter and I is the identity matrix.

In this study, in order to get real and imaginary images, the impedance was divided into two components, namely resistance components and capacitive reactance components. Equation (10) connected the impedance with the resistance components and the capacitive reactance components. In order to get a real image, Equation (10) becomes Equation (11):(11)$${\left[{\delta \rho }_{R}\right]}_{\left(q^{2}x1\right)}={\left({\left[S\right]}^{T}\left[S\right]+\alpha I\right)}^{-1}{}_{\left(\mathrm{pxp}\right)}{\left[S\right]}^{T}{}_{\left({\mathrm{pxq}}^{2}\right)}{\left[{\delta V}_{R}\right]}_{\left(\mathrm{pxp}\right)}$$

Equation (10) connected the impedance to the capacitive reactance components. In order to get an imaginary image, Equation (10) becomes Equation (12):(12)$${\left[{\delta \rho }_{X_{C}}\right]}_{\left(q^{2}x1\right)}={\left({\left[S\right]}^{T}\left[S\right]+\alpha I\right)}^{-1}{}_{\left(\mathrm{pxp}\right)}{\left[S\right]}^{T}{}_{\left({\mathrm{pxq}}^{2}\right)}{\left[{\delta V}_{\mathrm{Xc}}\right]}_{\left(\mathrm{pxp}\right)}$$

## 3. Results

The influence variables from the EIT system, as depicted in the Figure 3, consist of the stability of the Voltage Controlled Current Source (VCCS), the accuracy of the voltage between the electrodes, and the phase difference. The instability of the current will lead into invalid voltage measurements which can affect the obtained impedance value. The invalid impedance will directly affect the image of the reconstruction results. The AD 8302 phase detector is needed to measure the phase difference between the voltage on the object and the injection current. The phase difference in this experiment is required to determine the impedance components separately, which are resistive reactance and capacitive reactance. In addition, invalid phase difference measurement will affect the accuracy of the calculation result of resistive reactance and capacitive reactance. It will then directly influence the image of the reconstruction results. Overall, the final results of the EIT system are greatly influenced by the performance of VCCS and phase detector.

### 3.1. Voltage Controlled Current Source (VCCS) 

The VCCS performance can be seen from the stability of the currents to the electrical loads and the width of the frequency. Figure 9 shows a graph of the current stability of the experimental results of VCCS test. The VCCS produces a constant electric current of 1 mA that is stable with a load of 1 kΩ at frequencies up to 150 kHz. At the frequencies of 150–200 kHz, there is a 1% decrease in amplitude from the stable state. The 2 kΩ load on VCCS is stable until 100 kHz while for the 3 kΩ load reaches 80 kHz.

The effect of VCCS loads of 1–5 kΩ on the stability of frequency currents of 10–100 kHz is shown in Figure 8. The change in current value is quite small due to changes in loads. The stability of the current against the load is used to determine the suitability of the current source with the object to be detected. Anomalies in objects could be detected properly if the object impedance and anomalies are in the loads area of current source stability with the condition that in electrical impedance systems, anomalies and objects have different impedances. Meanwhile, the function of the current stability to the frequency changes is to measure the change in impedance when there is a change in the frequency of injection current. One of these applications is for the breast tumor detection system. There is a change in the characteristics of electrical conductivity of the tissue for different frequencies. Conductivity and impedance in normal tissue are constant, while conductivity and impedance of breast tumors change significantly at the current injection of 10 kHz and 100 kHz [7].

### 3.2. Voltage and Phase Difference Measurements 

The circuit scheme used to measure the voltage and the phase difference of this EIT is shown in Figure 7. This circuit has two functions, namely to measure the voltage of the object and to measure the phase difference between the injection current and the potential produced. Therefore, there are two mechanisms used in the testing. The first mechanism is the stability by measuring voltage at *V*_rms_ to DC. This test was done to ensure the accuracy of the voltage measurement. The stability was observed at the input and output voltages of 10, 20, 50, 100 mV at frequencies of 10 Hz to 550 kHz. The graph stability of the relation between the input voltage, the output voltage, and the frequency are displayed in Figure 10.

Figure 10 is a difference of *V_rms_* from DC output at 100 kHz to a lower frequency. The difference is very obvious, especially for the amplitude of 100 mV. In addition, it could be seen that the higher the input amplitude of *V_rms_* to DC, the bigger the difference in output voltage to the input voltage, which causes the measurement of same voltages at 100 kHz and at a lower frequency to be different. There must be a solution to solve the difference in the results of voltage measurements for the same input voltage at 100 kHz with a low frequency. In the process of image reconstruction, the problem of differences in measurement has been overcome by using the normalization method.

Related to the measurement of phase difference, the second activity is the linearity test of different phase conversion voltages by inputting the AD 8302 phase detector circuit. The aim of the test is to prove the feasibility of the circuit as a part of EIT system that acts to measure the phase difference. From the detector phase test, two types of data were obtained which show the feasibility of using the circuit. The first data were obtained from testing using two sinusoidal signals with the same amplitude and frequency, but there was the phase difference between the two signals that can be modified repeatedly. The same treatment for this test for sinusoidal signals’ frequencies at 1 kHz, 3 kHz, and 5 kHz.

The test results data are presented in graph which shows the relationship between the phase differences of the two input signals of AD 8302 phase detector to the output voltage (see Figure 11). This research can only prove the linearity of the phase difference to the output voltage up 5 kHz due to the limitations of the tools. Figure 10 shows that there is no change in the linearity graph due to changes in frequency. However, the results of this verification are the same as the IC sheet data. In the data sheet, it is explained that the IC AD8302 has the ability to detect changes of 10 mV/degree. This IC phase detector is capable of reading stable phase differences up to frequency signals of 2 MHz.

The second experiment was related to the phase detector, this second data were obtained by providing input of two sinusoidal signals with different amplitudes, same frequency, and different phase variations. The second experimental setup in Figure 8 was used to test the phase detector linearity which utilized two input of the sinusoidal signals (1 and 2) at different amplitudes with a ratio of V_1_/V_2_ = 1:1, 1:2, 1:3 and 1:4. Signal 1 and signal 2 were generated at 3 kHz. From these results, a linearity graph of the output voltage to the phase with the amplitude ratio mentioned was created (see Figure 11). The graph explains the plotting result of the second experiment. The graph shows the same linearity even though the two input amplitudes are different. This linear graph also illustrates the difference in amplitude of the two signals would shift the linearity position but does not change the linearity between the phase difference and the output voltage. The shift in the position of the linearity gives the constant information that changes due to changes in the amplitude ratio. Data related to the linearity characteristics of Figure 11 are shown in Table 1.

From the graph of the linearity data in Table 1, a plot of the relationship between the constants and the ratio V_1_/V_2_ was made. The relationship between the ratios of the two input signal amplitudes to the constants of the linearity shifting is shown in Figure 12. The graph shows that the relationship between the amplitude ratio and the shift constant is linear. The linear function of the relationship will provide a solution for the linearity shift in the phase detector. If the amplitude ratio of V_1_ and V_2_ has been obtained, then from the linear equation in Figure 12, the constant value information can be acquired. Thereafter, the linier graph of the relationship between the phase detector output voltage and the phase difference was generated. Furthermore, from the phase detector output voltage, the phase from the two signals are obtained. This step is used to measure the phase difference between voltage and current injection with AD 8302 phase detector.

### 3.3. Simulation

Simulations are carried out to solve forward problems using Finite Element Methods (FEM) so that potential data is obtained from numerical objects. The numerical object is built from the electrical properties of the heart and deflated lungs at Table 2. The simulation uses the neighbor data collection method. The triangular element model used in this study consists of 141 nodes, 248 elements, and 16 electrodes, as illustrated in Figure 13.

From Figure 14a–c and Figure 15a–c, it can be seen that the heart and deflated lung have the same impedance, but different real and imaginary images, respectively.

### 3.4. Image Scanning and Reconstruction

In this stage, we integrated several devices that had been built and the available modules. This integration process produced a system of electrical impedance metallography (EIT), as shown in Figure 4. The scanning process was carried out by using the neighboring method to obtain electric potential data and the phase difference between the current injection current and the measured potential. This EIT system was used to scan phantom objects.

The phantom object is a circle container filled with water as a medium and a carrot as an anomaly, as depicted in Figure 16. Data from Tusarkanti Bera [5] shows that the carrot object at the frequency of 10 kHz has a Z value of 7.2 kΩ, real 6.2 kΩ, and imaginary = −3.2 kΩ. Whereas at the frequency of 100 kHz it has a value of Z = 2.2 kΩ, real = 1.3 kΩ, and imaginary = −1.5 kΩ. The electrodes made of printed circuit board PCB were attached around the edge of the phantom object. The scanning process used AC current injection at 10 and 100 kHz. From the scanning process, two data were obtained, namely the electric potential and the phase difference. The electric potential data and the difference in the results of the scanning process are shown in Figure 17. We can observe that there are potential differences measured at different frequencies. This potential graph demonstrates a periodic pattern of measurements on all electrodes and in the pairs of electrode injection. In the phase difference pattern (Figure 17) for the 100 kHz, it depicts the correct pattern but not quite smooth due to the imperfect tool, so that needs to be improved in the future. 

From the multiplication of the electric potential and the phase difference data from Figure 17 and by using Equations (3) and (4), real potential and imaginary potential were obtained for each corresponding value, namely [V_Z_], [V_Cos*φ*_], [V_sin*φ*_]. The reconstruction of the three potentials was conducted to get the images. From the reconstruction process data impedance, resistive reactance, and capacitive were obtained. This reconstruction process uses a linear method. All three data were taken from two different frequencies. Thus, the resulting image is a functional image. The image of the reconstruction results from the impedance data is called the impedance image while the reconstruction image of the resistive reactance data is called the real image, and the reconstructed image of the capacitive reactance data is called the imaginary image.

The experiments obtained functional images for impedance, resistive (real), and capacitive (imaginary). By using α = 0.01, potential and phase data are reconstructed to produce impedance images, real images, and imaginary images as shown in Figure 18. These three images increase the accuracy of the EIT system in determining the abnormality of objects that have the same impedance. The statistical analysis used to show image quality is the standard deviation. It calculates the relative errors of the impedance, real, and imaginary images. The value of each impedance, real, and imaginary image are 0.01621, 0.02476, and 0.02123, respectively.

## 4. Discussion

The noninverting op-amp added at the VCCS double op-amp [27] AD8039 can increase the current. VCCS performance is displayed in the frequency response graph. VCCS with a given load of 3 kΩ generates a steady current of 0.1–1.5 mA at frequencies 500 Hz–80 kHz, at a frequency of 90 kHz current drops 0.03%, and at a frequency of 100 kHz the current drops 0.05% from a stable state. VCCS with a load of 2 kΩ produces a steady current of 0.1–1.5 mA for at a frequency of 100 kHz. The current used in this experiment is 1 mA. The current used on the EIT depends on the object. The current generated by VCCS depends on the magnitude of the gain.

This research requires two currents with a frequency of 10 kHz and 100 kHz. The dual frequencies appertain to the method used to detect abnormalities, which is using functional imagery [28,29]. However, this VCCS can produce stable currents for a wide frequency. VCCS double op-amps are capable to support multi-frequency and programmable EIT. As in the Mf-EIT study, the required frequencies are 5 10, 25, 50, 75, and 100 kHz [5]. This VCCS can generate a stable current at those frequencies.

This EIT has succeeded in separating the impedance distribution into two namely the resistance distribution and capacitive reactance. The method used to separate these components is indeed simple, which is by using the analogy of a series of RC series with body tissue. By measuring the potential, the phase difference between the potential and the injected current, “forward problem” and “inverse problem”, we can obtain the distribution of impedance, resistance, and capacitive reactance.

Distribution of the three components obtained three images, namely the impedance image, resistance, and capacitive reactance. The resistance image is taken from the real impedance component so it is called the real image, while the capacitive reactance image is taken from the imaginary impedance component so it is called an imaginary image.

The purpose of the separation of two components is to detect the material specifications of an object in a more detailed manner. This research is a preliminary result to distinguish inhomogeneous material. From the three images, it can be seen that there are differences in the image profiles of the same object. Each material or network has an identical image impedance, resistance, and reactance image for each tissue. These three images have improved the accuracy of the EIT system in detecting abnormalities when compared to just one image [2,3,4,5,6,7,8,9,10,11,16]. The results obtained have been compared with data from other researchers’ experiments [3], the results of these comparisons show a corresponding trend. That the Z image is bigger than the real image and imaginary valued negative. Which is all in accordance with the carrot impedance values both Z, real, and imaginary values.

Algorithms and data acquisition systems to separate resistance and capacitive reactance have been successfully developed. Although EIT has a problem in image resolution, the LBP and Tikhonov methods in EIT can produce high contrast images. The reference reconstruction method requires reference data that can be obtained from different frequency data so that this reconstruction method requires two data from different frequencies. The reconstruction produces a relative image or also called a functional image.

Jiang and Soleimani [12] have developed Capacitively Coupled Electrical impedance Tomography CCEIT for brain imaging. Whereas Wang et al. have developed a suitable reconstruction method for the CCEIT system using fusion images. Real, imaginary, and magnitude images from the LBP method are summed and by using K-means as the filter threshold, a better image has been obtained [30]. The advantage of the CCEIT system is that it requires no contact between the electrode and the object, thus the system is safer when compared to a standard EIT. The CCEIT developed by Wang [30] uses a total impedance of two-phase gas-liquid flow. This research produces real, imaginary, and impedance images, and combines real and imaginary parts. Images reconstructed from various impedances are investigated and discussed. The LBP reconstruction method is used to obtain object images from various impedance parts and K-means are adopted to obtain the gray level threshold automatically. The weighted combination method was introduced to combine images by combining images reconstructed from real and imaginary impedances.

Our research obtains impedance, real, and imaginary images. All images are obtained from potential data and phases that can be collected from the developed system. LBP reconstruction method from two different frequency data can produce the three images. This is the similarity between our research and CCEIT. The advantage of CCEIT compared to our study is that it is non-contact, so it can be used for objects that have high resistivity on the surface. 

Studies by Jiang and Wang [12,30] stated that the suitable object for CCEIT applications is head. One of the requirements in using electrical modality to observe the brain is that it should be able to penetrate the skull that have high resistivity therefore the EIT system is less suitable for this use. However, for organs that do not have high resistivity such as the thorax and abdomen, the EIT system is suitable. The system that we have developed can be used for imaging organs in the thorax or abdomen in the future. As we explained earlier that there are organs in the thorax and abdomen that have the same impedance but with a different conductivity and permittivity. The EIT system that we have developed can produce images not only impedance but also of real and imaginary images. The last two images are embodiments of the conductivity and permittivity of the object.

The study has conducted a design and development EIT device without involving a high-speed data acquisition system but it can produce impedance image components that include real and imaginary images. The EIT provides important information that tissue impedance components have been successfully separated. These three images show a more detailed and specific identity of a tissue. Three images are more accurate in distinguishing different objects with the same impedance composed of different impedance components.

## 5. Conclusions

This research has succeeded to develop the EIT system that can separate impedance components. The impedance components consist of resistive reactance as the real parts and capacitive reactance as the imaginary parts This experiment utilized alternating current with frequencies of 10 and 100 kHz. Two current sources from different frequencies was injected and produce voltage and phase data. Two current sources from different frequencies was injected and produce voltage and phase data, The linear back projection was used to reconstruct voltage and phase data to generate impedance, real and imaginary images. It can be concluded that the EIT system can produce functional impedance, real and imaginary images. The functional real and imaginary images more sensitive in detecting anomalies compared to the functional impedance image. It can be seen from the capacitive reactance image of carrots as an anomaly in the water that looks clearer than in the impedance functional image. The anomaly in the resistive reactance image is still not visible which may be because the resistance of carrots is worth the same as the water medium.

## Figures and Tables

**Figure 1 sensors-20-01907-f001:**
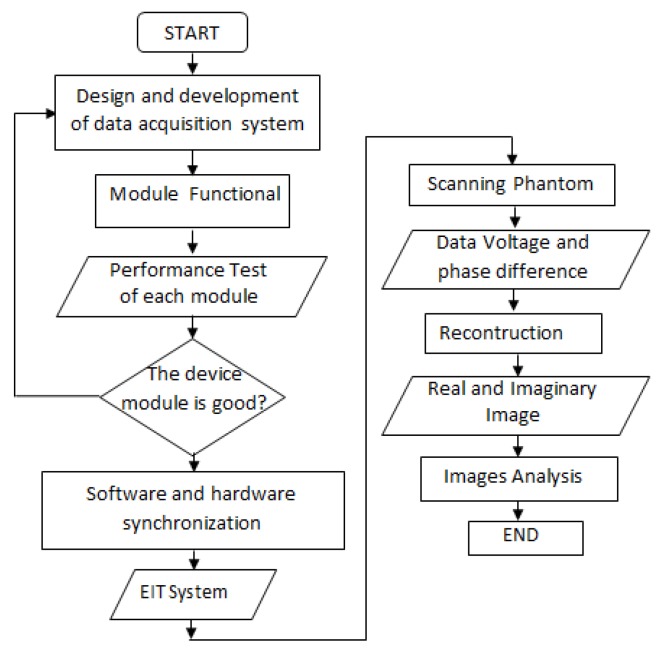
Schematic diagram of the experimental procedure.

**Figure 2 sensors-20-01907-f002:**
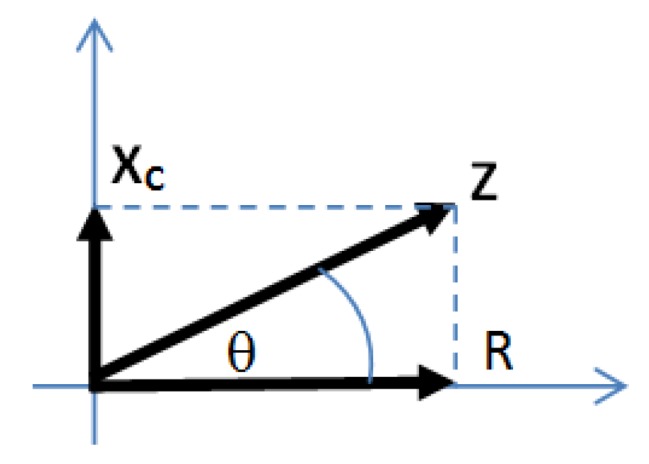
The relation of R, Xc, and Z.

**Figure 3 sensors-20-01907-f003:**
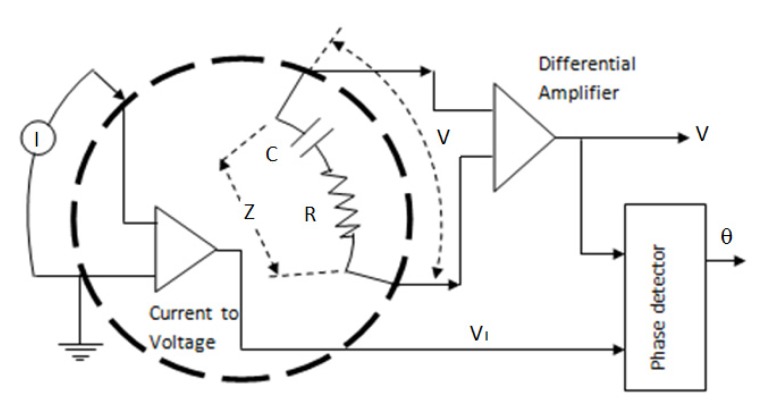
The scheme in the measurement of impedance components separately through a measurement of voltage and phase difference.

**Figure 4 sensors-20-01907-f004:**
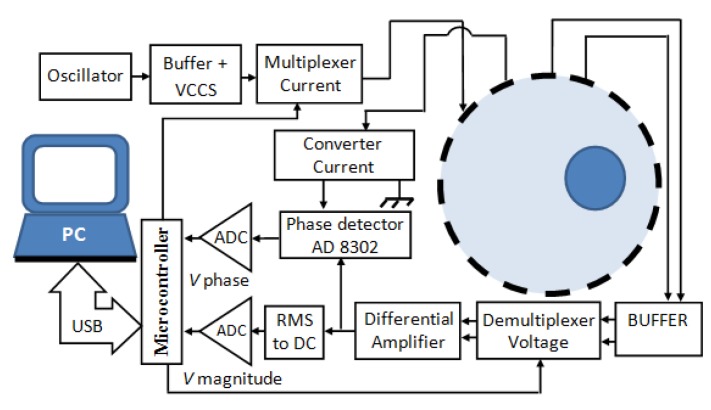
Block diagram of Electric Impedance Tomography System for real and imaginary functional images.

**Figure 5 sensors-20-01907-f005:**
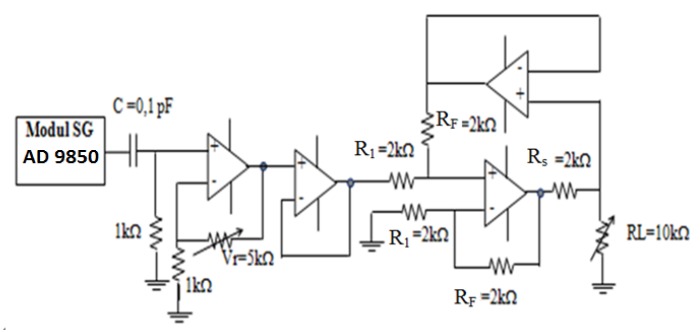
A signal generator and constant current.

**Figure 6 sensors-20-01907-f006:**
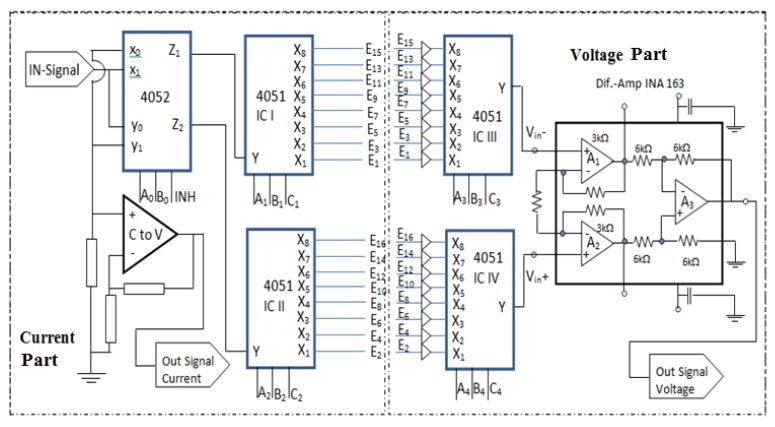
The electric mechanism for selecting a current injector electrode and measuring voltage.

**Figure 7 sensors-20-01907-f007:**
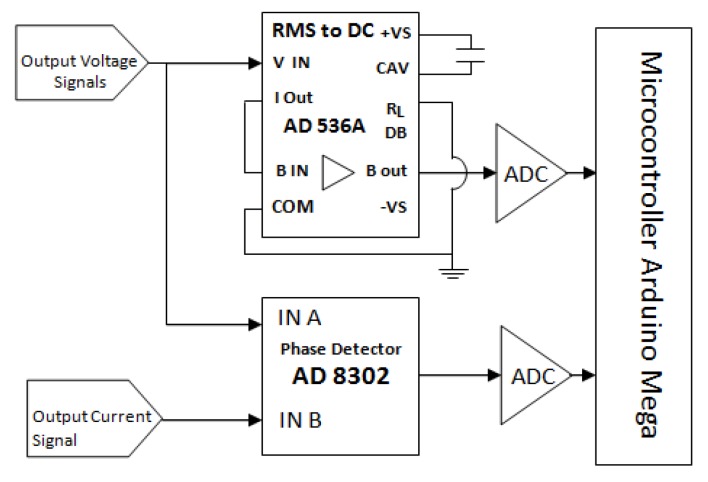
Measurement scheme of voltage and phase difference.

**Figure 8 sensors-20-01907-f008:**
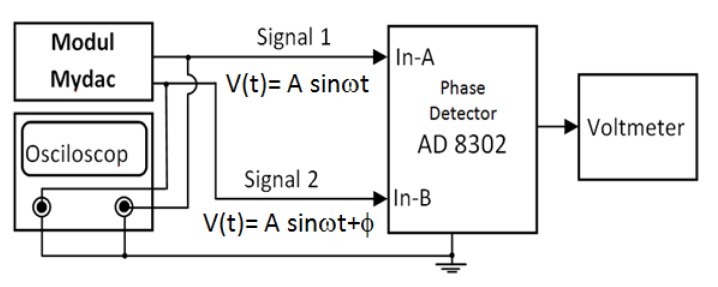
The set up for the linearity test of Phase Detector circuit.

**Figure 9 sensors-20-01907-f009:**
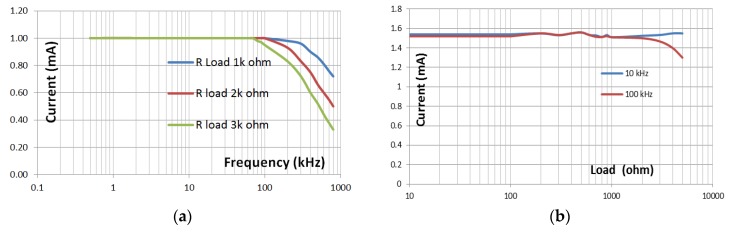
(**a**) The relationship between frequencies and loads in the Voltage Controlled Current Source (VCCS) to the currents; (**b**) the graphic of current stability to loads at 10 kHz and 100 kHz.

**Figure 10 sensors-20-01907-f010:**
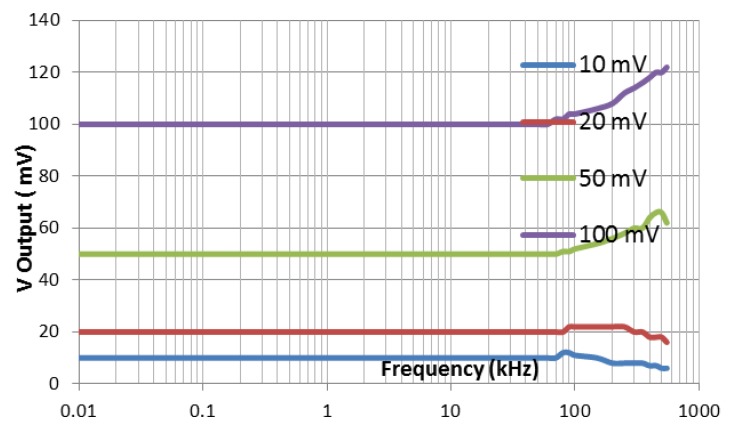
The stability of root mean square (RMS) to direct current (DC) circuit of output voltage to DC against frequencies.

**Figure 11 sensors-20-01907-f011:**
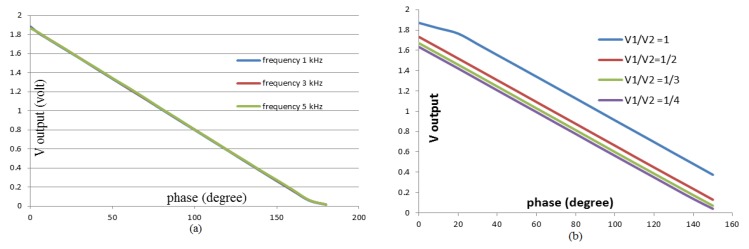
(**a**) The relation of the linearity of phase differences and voltages from signals detector with the same amplitude. (**b**) The linearity graph of phase detector’s voltages to the phase differences for two different signal amplitudes.

**Figure 12 sensors-20-01907-f012:**
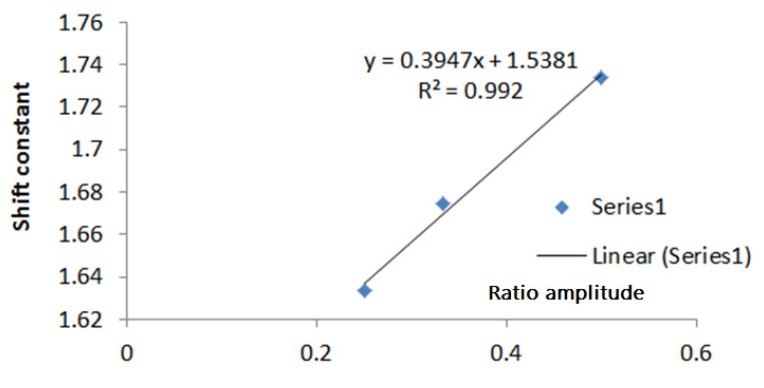
The relation of the shift constants and the amplitude ratio.

**Figure 13 sensors-20-01907-f013:**
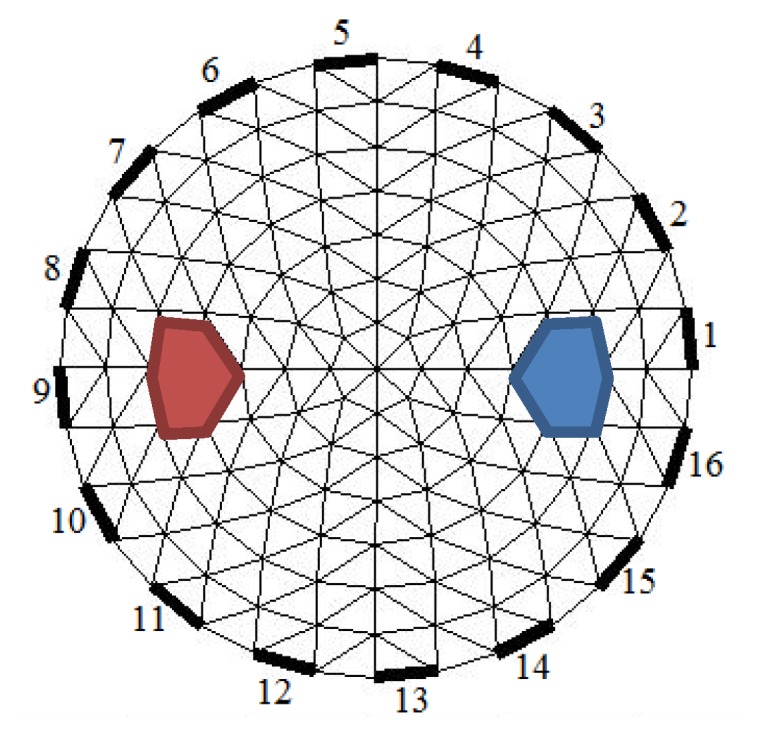
The triangular element model used in this study.

**Figure 14 sensors-20-01907-f014:**
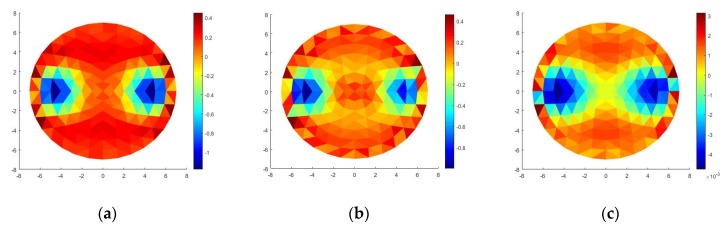
(**a**) Impedance image, (**b**) real image, (**c**) imaginary image.

**Figure 15 sensors-20-01907-f015:**
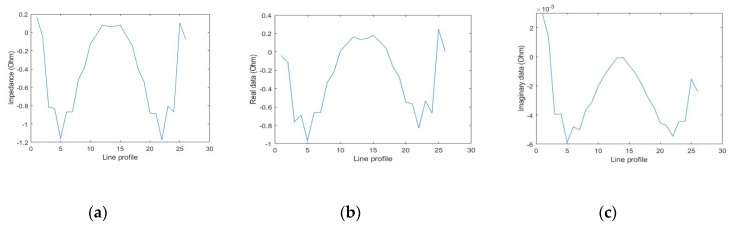
Line profile of (**a**) impedance image, (**b**) real image, (**c**) imaginary image.

**Figure 16 sensors-20-01907-f016:**
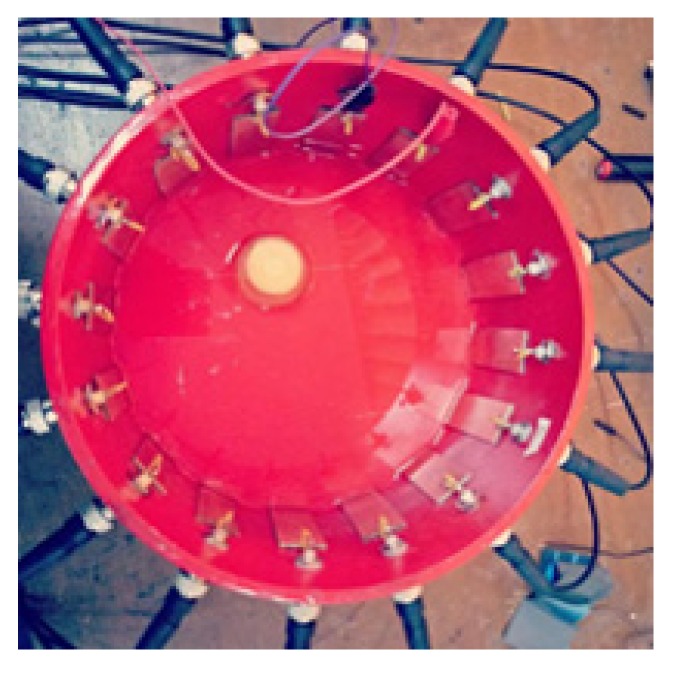
Phantom object.

**Figure 17 sensors-20-01907-f017:**
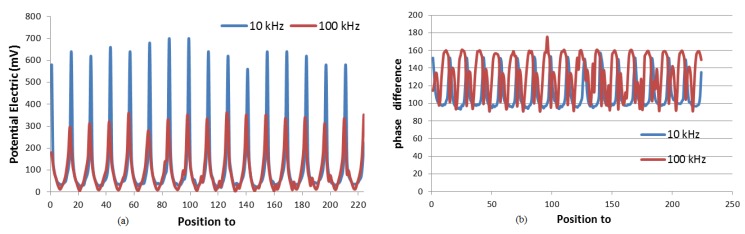
(**a**) Data of electric potential from the scanning using Neighboring method. (**b**) Data of phase difference from the scanning using Neighboring method.

**Figure 18 sensors-20-01907-f018:**
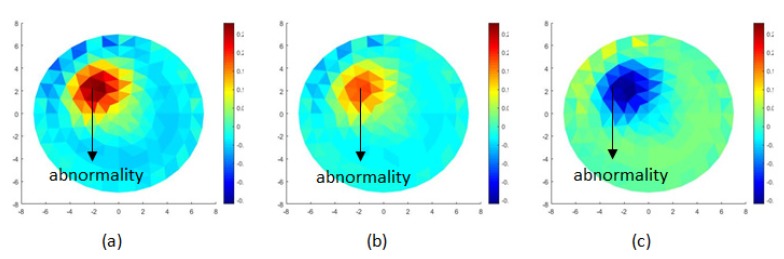
(**a**) Impedance functional image, (**b**) real functional image, (**c**) imaginary functional image.

**Table 1 sensors-20-01907-t001:** Data of the linearity shift and the phase detector’s ratio of 2 signal amplitudes.

No	Ratio V_1_/V_2_	Constant	Slope
1	1	1.9455	0.0104
2	½	1.7338	0.0107
3	1/3	1.6747	0.0107
4	1/4	1.6334	0.0107

**Table 2 sensors-20-01907-t002:** Data of the electrical properties many tissue from Gabriel.

		Frequency				
	10 kHz			100 kHz		
	Heart	Deflated Lung	Distillated Water	Heart	Deflated Lung	Distillated Water
Z (Ω)	1.001	1.078	2000	2.676	2.988	2000
Re (Ω)	0.245	0.084	2000	0.66	0.313	2000
Im (Ω)	0.971	1.074	100	2.593	2.972	100

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
