# Peer review of "Anomaly Detection Using Electric Impedance Tomography Based on Real and Imaginary Images"

_sensors, 2020, doi:10.3390/s20071907_

Round 1
Reviewer 1 Report
The paper described the development of a dual-frequency EIT system. The topic is interesting. But the paper is badly organized. Some improved issues should be decreased.
(1) Please highlight the novelty of the proposed system. After all, there are already plenty of papers on this topic.
(2) The MF-EIT is efficient in the clinical applications due to its wind frequency scanning band could improve its sensitivity to different tissue. The advantage of the proposed method on MF-EIT is fully wrong.
(1) It seems like the difference imaging is used. Please describe the method of selecting the reference distribution.
(2) According to the description, S in equation (7) is calculated with the perturbation method. What is the physical meaning of the S?
(3) Why the LBP is described before the Tikhonov method? The LBP seems like has not been used in the paper.
(4) The phantom can not be distinguished from Fig. 17. According to this result, one can not see the system is working.
Author Response
Reviewer # 1 :
The paper described the development of a dual-frequency EIT system. The topic is interesting. But the paper is badly organized. Some improved issues should be decreased.
- Please highlight the novelty of the proposed system. After all, there are already plenty of papers on this topic.
Ans : in the last background we have add the sentence:
The study have conducted design and development EIT device without involving a high-speed data acquisition system but it can produce impedance image components that include real and imaginary images
(2) The MF-EIT is efficient in the clinical applications due to its wind frequency scanning band could improve its sensitivity to different tissue. The advantage of the proposed method on MF-EIT is fully wrong.
Ans: We have removed the confusing sentence:
The image of the reconstruction results starts to blur for 200 kHz. For the frequencies above 300 kHz, the image is not obtained [8] which is resulted from the decrease in the resistivity of potatoes based on the impedance spectrum value at different current frequencies.
- It seems like the difference imaging is used. Please describe the method of selecting the reference distribution.
Ans :
The reference reconstruction method requires reference data that can be obtained from different frequency data so that this reconstruction method requires two data from different frequencies.
(2) According to the description, S in equation (7) is calculated with the perturbation method. What is the physical meaning of the S?
Ans : We have decleared with the sentence :
Generally, is a singular matrix so that the matrix does not have inverse. To solve these problems, Tikonov regulation can be used so that the matrix has inverse
(3) Why the LBP is described before the Tikhonov method? The LBP seems like has not been used in the paper.
Ans : Tikhonov is only used to complete the LBP reconstruction method so that matrices that are originally singular become non-singular so there is a solution.
(4) The phantom can not be distinguished from Fig. 17. According to this result, one can not see the system is working.
Ans : We have re-experimented with a better image of reconstruction.

Reviewer 2 Report
This paper is about Abnormality Detection Using EIT separatind the distribution of two impedance components: the resistive component and the capacitive reactance component. The present study uses different impedances with normal and pathological tissues for detecting abnormalities.
The following are the comments in which authors should be address.
1) Lines 43, 44 and 45: more articles of using AI should be cited;
2) Line 51: the word “electricalal” should be corrected;
3) Page 3, lines 103-116: the diagram is too smaill;
4) Lines 126-129 and 134-136: the equations should be centered;
5) Figure 3: the texts on it are too small;
6) Lines 70-71: it should be better explained what is the criteria of inserting the values of R and C for the equivalent circuit;
7) The Neighboring method should be better explained;
8) Lines 89-91: The explanation of what is functional and diferential imagens is repetitive, as they have already explained in lines 84-88;
9) The diference of functional and diferential images are very important in this article, then a better explanation should be considered by the authors, as for example the one find in the work of Borvik;
10) Lines 119-136: the explanation would be better addressed by adding a plot of resistance versus reactance, showing the theta angle;
11) Line 313: Why was the method Filtered Backprojection chosen? Are not there others more effetive? Has not already proven that the C method does not produce trustuble images?
12) Line 313: the word "Filtere" should be corrected;
13) Line 316: the references Barber & Brown should be cited together with the [21];
14) Line 332: the word "Howevver" should be corrected;
15) Line 337: the correct name for this melhod is "Tikhonov regularization";
16) Line 385: the word "realtion" should be corrected;
17) Figure 13: the text "rasio" inside the figure should be corrected;
18) Lines 517-529: the text described here does not reflect a discussion but the concept of EIT instead;
19) The results obtained here were not discussed;
20) Lines 540-541: the authors say that the imaging reconstrution methods were succefully used. This cannot be possible according to literature: Filtered Backprojection e Tikhonov regularization have important issues not yet overcome in the production of reliable images.
Author Response
Reviewer # 2 :
This paper is about Abnormality Detection Using EIT separatind the distribution of two impedance components: the resistive component and the capacitive reactance component. The present study uses different impedances with normal and pathological tissues for detecting abnormalities.
The following are the comments in which authors should be address.
1) Lines 43, 44 and 45: more articles of using AI should be cited;
Ans: Has been deleted
2) Line 51: the word “electricalal” should be corrected;
Ans: Done
3) Page 3, lines 103-116: the diagram is too smaill;
Ans ; Done
4) Lines 126-129 and 134-136: the equations should be centered;
Ans ; Done
5) Figure 3: the texts on it are too small;
Ans ; Done
6) Lines 70-71: it should be better explained what is the criteria of inserting the values of R and C for the equivalent circuit;
Ans:
All series and parallel circuits consisting of resistors R and capacitors C can be replaced with a value equivalent to a single RC series
Replace by :
All series and or parallel circuits consisting of resistors R and capacitors C heve an aquivalent to an RC series. The circuit consists of RC components in series and or parallel relationships, regardless of the number and value of RC can be simplified in the final form of RC series with a certain value.
7) The Neighboring method should be better explained;
Ans :
The measurement method used is the Neighboring method
Replace by :
The study used the Neighboring data collection method with 16 electrodes. This method works by injecting current and measuring the voltage between the two closest electrodes.
8) Lines 89-91: The explanation of what is functional and diferential imagens is repetitive, as they have already explained in lines 84-88;
Ans :repetitive has been deleted
9) The diference of functional and diferential images are very important in this article, then a better explanation should be considered by the authors, as for example the one find in the work of Borvik;
Ans: The reference reconstruction method requires reference data that can be obtained from different frequency data so that this reconstruction method requires two data from different frequencies.
10) Lines 119-136: the explanation would be better addressed by adding a plot of resistance versus
reactance, showing the theta angle;
Ans: The relation of R, Xc and Z is shown in Figure 2
11) Line 313: Why was the method Filtered Backprojection chosen? Are not there others more effetive? Has not already proven that the C method does not produce trustuble images?
Ans:
12) Line 313: the word "Filtere" should be corrected;
Ans : Done
13) Line 316: the references Barber & Brown should be cited together with the [21];
Ans : Done
14) Line 332: the word "Howevver" should be corrected;
Ans : Done
15) Line 337: the correct name for this melhod is "Tikhonov regularization";
Ans : Done
16) Line 385: the word "realtion" should be corrected;
Ans : Done
17) Figure 13: the text "rasio" inside the figure should be corrected;
Ans : Done
18) Lines 517-529: the text described here does not reflect a discussion but the concept of EIT instead;
Ans: This paragraph emphasizes the success of the research that has been carried out.
19) The results obtained here were not discussed;
Ans: in the Line 479-199 we have added with a discussion of the results
The separation of two components is to detect in more detail the material specifications of an object. This research is a preliminary result to distinguish inhomogeneous material. From the three images, it can be seen that there are differences in the image profiles of the same object. Each material or network has an identical image impedance, resistance and reactance image for each network. These three images have improved the accuracy of the EIT system in detecting abnormalities when compared to just one image [2-11, 16];. The results obtained have been compared with data from other researchers' experiments [3], the results of these comparisons show a corresponding trend. That the Z image is bigger than the real image and imaginary valued negative. Which is all in accordance with the carrot impedance values both Z, Real and Imaginary values.
Algorithms and data acquisition systems to separate resistance and capacitive reactance have been successfully carried out. Although EIT has a problem in image resolution, the LBP and Tikhonov methods in EIT can produce high contrast images. The reference reconstruction method requires reference data that can be obtained from different frequency data so that this reconstruction method requires two data from different frequencies. The reconstruction produces a relative image or also called a functional images.
The study have conducted design and development EIT device without involving a high-speed data acquisition system but it can produce impedance image components that include real and imaginary images. The EIT provides important information that tissue impedance components have been successfully separated. These three images show a more detailed and specific identity of a tissue. Three images are more accurate in distinguishing different objects with the same impedance composed of different impedance components.
20) Lines 540-541: the authors say that the imaging reconstrution methods were succefully used. This cannot be possible according to literature: Filtered Backprojection e Tikhonov regularization have important issues not yet overcome in the production of reliable images.
Ans:
Although EIT has a problem in image resolution, the FBP and Tikhonov methods in EIT can produce high contrast images. This image can be used to produce functional images.

Reviewer 3 Report
This work considers the problem of abnormality detection using EIT technology. Both the real and the imaginary images are utilized to improve accuracy in determining object abnormalities. The components of network impedance are separated by analogizing body tissue with a series of RC series. Phantom experiments are conducted to verify the effectiveness of the proposed method.
- There are a lot of typos and non-technical errors in this manuscript. For instance, one should not use "The" in the title; researchers commonly term the reconstruction process of EIT images "inverse problem", not "reverse problem"; Page 7, Line 248, the reduntant "i" in the word "Electroniic"; Page 9, Line 313, "The Image reconstruction was done by using the reconstruction method of Filtere Back Projection." This sentence alone has at least two typos: the capitalized "I" in the word "Image" and the missing "d" in "Filtere". This manuscript contains way too many such problems that I will not be in a position to fix.
- Please improve the image quality of the figures in this paper. For example, Fig. 1-3 are too small for anyone to read the text in the figure. Just show photographs as in Fig. 14 is not informative--annotations are required. Some related figure pairs such as Fig. 8-9, Fig. 11-12, Fig. 15-16 should be merged into one figure with two subfigures.
- The experimental validation is superficial, and no comparison is presented, which makes the conclusion unconvincing. Just showing results with such a simple experimental setting is insufficient. Statistical analyses with respect to various quantitative and objective metrics should also be included.
- Many relevent and important references are ignored in the discussion. For instance:
[Ref1] "Capacitively Coupled Electrical Impedance Tomography for Brain Imaging", IEEE Transactions on Medical Imaging, (DOI: 10.1109/TMI.2019.2895035).
[Ref2] "Efficient multi-task structure-aware sparse Bayesian learning for frequency-difference electrical impedance tomography", IEEE Transactions on Industrial Informatics, (DOI: 10.1109/TII.2020.2965202).
[Ref3] "Study on image reconstruction of capacitively coupled electrical impedance tomography (CCEIT)", Measurement Science and Technology, (DOI: 10.1088/1361-6501/ab1324).
Author Response
Reviewer #3
This work considers the problem of abnormality detection using EIT technology. Both the real and the imaginary images are utilized to improve accuracy in determining object abnormalities. The components of network impedance are separated by analogizing body tissue with a series of RC series. Phantom experiments are conducted to verify the effectiveness of the proposed method.
- There are a lot of typos and non-technical errors in this manuscript. For instance, one should not use "The" in the title; researchers commonly term the reconstruction process of EIT images "inverse problem", not "reverse problem"; Page 7, Line 248, the reduntant "i" in the word "Electroniic"; Page 9, Line 313, "The Image reconstruction was done by using the reconstruction method of Filtere Back Projection." This sentence alone has at least two typos: the capitalized "I" in the word "Image" and the missing "d" in "Filtere". This manuscript contains way too many such problems that I will not be in a position to fix.
*should not use "The" in the title
Ans : The title replace by
Detection of Abnormalities Using Electric Impedance Tomography Based on Analysis of Real and Imaginary Images
*"inverse problem", not "reverse problem"
Ans: "reverse problem " has been replace by “inverse problem"
*Page 7, Line 248, the reduntant "i" in the word "Electroniic";
Ans : Done
Page 9, Line 313, "The Image reconstruction was done by using the reconstruction method of Filtere Back Projection." This sentence alone has at least two typos: the capitalized "I" in the word "Image" and the missing "d" in "Filtere".
Ans : Done
- Please improve the image quality of the figures in this paper. For example, Fig. 1-3 are too small for anyone to read the text in the figure. Just show photographs as in Fig. 14 is not informative--annotations are required. Some related figure pairs such as Fig. 8-9, Fig. 11-12, Fig. 15-16 should be merged into one figure with two subfigures.
* Fig. 1-3 are too small for anyone to read the text in the figure
Ans : Fig 1-3 has been fixed
*Just show photographs as in Fig. 14 is not informative--annotations are required.
Ans : in the line 484-487 we have add the Sentence:
Data from Tusarkanti Bera [5] shows that the carrot object for a frequency of 10 kHz has a Z value of 7.2 kW, Real 6.2 kW and Imaginary= -3.2 kW. Whereas for a frequency of 100 kHz has a value of Z = 2.2 kW, Real = 1.3 kW and Imaginary = -1.5 kW.
*Some related figure pairs such as Fig. 8-9, Fig. 11-12, Fig. 15-16 should be merged into one figure with two subfigures.
Ans : Fig. 8-9, Fig. 11-12, Fig. 15-16 has been fixed
- The experimental validation is superficial, and no comparison is presented, which makes the conclusion unconvincing. Just showing results with such a simple experimental setting is insufficient. Statistical analyses with respect to various quantitative and objective metrics should also be included.
Ans:
The results obtained have been compared with data from other researchers' experiments (Bera 2016), the results of these comparisons show a corresponding trend. That the Z image is bigger than the real image and imaginary valued negative. Which is all in accordance with the carrot impedance values both Z, Real and Imaginary values.
- Many relevent and important references are ignored in the discussion. For instance:
[Ref1] "Capacitively Coupled Electrical Impedance Tomography for Brain Imaging", IEEE Transactions on Medical Imaging, (DOI: 10.1109/TMI.2019.2895035).
[Ref2] "Efficient multi-task structure-aware sparse Bayesian learning for frequency-difference electrical impedance tomography", IEEE Transactions on Industrial Informatics, (DOI: 10.1109/TII.2020.2965202).
[Ref3] "Study on image reconstruction of capacitively coupled electrical impedance tomography (CCEIT)", Measurement Science and Technology, (DOI: 10.1088/1361-6501/ab1324).

Reviewer 4 Report
This is a well written paper, which can be of interest for the readers in the field of impedance tomography. The authors want to get both - the real and imaginary images, but they use magnitude and phase measurements.
The first question: why do not measure the real and imaginary parts directly using the two-phase (inphase and quadrature) synchronous demodulation or Fourier transform?
The next question: what is new in the proposed solutions? I cannot see much in comparison with those of previous research, given for example, in the proceedings of regular Int. Conferences on Biomedical Application of Electrical Impedance Tomography (EIT). Already the 20th conference was held in 2019.
I guess that the authors should answer to the above given questions before publishing their manuscript.
Author Response
Reviewer # 4
This is a well written paper, which can be of interest for the readers in the field of impedance tomography. The authors want to get both - the real and imaginary images, but they use magnitude and phase measurements.
The first question: why do not measure the real and imaginary parts directly using the two-phase (inphase and quadrature) synchronous demodulation or Fourier transform?
Ans:
The real and imaginary part of this impedance can be measured using Fourier transforms. The method is simply, however we try to avoid data acquisition systems or high-speed sampling systems in this study.
The next question: what is new in the proposed solutions? I cannot see much in comparison with those of previous research, given for example, in the proceedings of regular Int. Conferences on Biomedical Application of Electrical Impedance Tomography (EIT). Already the 20th conference was held in 2019.
Ans :
I have add the sentences in line 84:
The study have conducted design and development EIT device without involving a high-speed data acquisition system but it can produce impedance image components that include real and imaginary images.
I guess that the authors should answer to the above given questions before publishing their manuscript.
I have add 2 reference from Int. Conferences on Biomedical Application of Electrical Impedance Tomography (EIT) 2019.

Round 2
Reviewer 1 Report
Thanks for answering the comments. The paper has been improved. The results are much better than the last version. However, the introduction section should be improved to make the novelty more clear.
(1) The author should double-check the introduction section to avoid misleading readers. For example, the expression in Page 2, line 57-60 is incorrect.
“same impedance include heart 47.74 Ω, kidney 47.95 Ω and gall 59 bladder 47.98 Ω. Impedance measurements of kidney and muscle at 10 GHz have almost the 60 same impedance value [23].”
Firstly, there are no applications of EIT on imaging the heart, kidney, and bladder at the same time. Actually, the EIT is barely used for kidney and bladder imaging. Secondly, the 10 GHz frequency is out of EIT range.
(2) A short review of the reported EIT system should be added, especially the system measuring the imaging and resistance components together.

Author Response
Reviewer 1
(1) The author should double-check the introduction section to avoid misleading readers. For example, the expression in Page 2, line 57-60 is incorrect.
“same impedance include heart 47.74 Ω, kidney 47.95 Ω and gall 59 bladder 47.98 Ω. Impedance measurements of kidney and muscle at 10 GHz have almost the 60 same impedance value [23].”
Firstly, there are no applications of EIT on imaging the heart, kidney, and bladder at the same time. Actually, the EIT is barely used for kidney and bladder imaging. Secondly, the 10 GHz frequency is out of EIT range.
Ans : Revisions have been conducted by replacing the impedance data of several objects at frequencies of 10 kHz and 100 kHz, obtained from reference C. Gabriel (1996)
The weakness of the impedance image generated from the MF-EIT system [8, 13] cannot distinguish objects that have the same impedance but different tissue. Some tissues that have almost the same impedance include liver and lungs have impedances of 1,622 and 1,624 W, heart and lung deflated have impedances of 1,001 and 1,077 W at a frequency of 10 Khz. While Colon and cervix have impedances of 3.3168 and 3.2875 W at a frequency of 100 kHz. The data was obtained from a simulation developed by C. Gabriel [22]. The impedance of these organs has different conductivity and permittivity although the impedance is almost the same. As a result, the Mf-EIT system cannot show differences in tissue that have the same impedance. However, if real and imaginary images are produced, they will be able to show the difference between the two objects which have the same impedance.
(2) A short review of the reported EIT system should be added, especially the system measuring the imaging and resistance components together.
Ans : We have write two pharagraph as short review about EIT system at line 77-95
The application of this method begins by assuming body tissue as a series of RC series. Furthermore, the body tissue is injected with AC current. At some point voltage and "phase difference" are measured. The study used the Neighboring data collection method with 16 electrodes. This method works by injecting current and measuring the voltage between the two closest electrodes [4]. The intended voltage is the difference in electrical potential at two specific points when current is injected into the object. Meanwhile, the phase difference is the phase difference between the voltage and the injection electric current. From these two data, impedance, resistive and capacitive reactance can be calculated. This concept is applied to build EIT in mapping the impedance, resistive and capacitive reactance of body tissues. In the reconstruction process the resistive reactance potential produces a real image, capacitive reactance produces a capacitive image, and the impedance produces an impedance image.
The study have conducted design and development EIT device without involving a high-speed data acquisition system but it can produce impedance image components that include real and imaginary images. To show anomalies in normal tissues, EIT utilizes a functional image [23]. The image uses two different frequencies of injection current [4, 28]. The research used current sources at frequency 10 kHz and 100 kHz. EIT produces three functional images, namely impedance image, resistive image and capacitive image. These three images can be used in the processes of identifying and analyzing tissues. Moreover, with these three images, tissue diagnostics and analysis are improved to be more accurate than by using only one impedance image.

Reviewer 3 Report
I do not think that my review comments have been adequately addressed -- some of them are even outright ignored. I believe the following issues MUST be fixed before any further considerations.
- The experimental validation is STILL superficial. The authors just added ONE sentence describing data from another totally different experiement as a "comparison", which is not convincing AT ALL!
- Again, just showing results with such a simple experimental setting -- one carrot cylinder in the water tank, is EXTREMELY insufficient. The authors made NO revisions with respect to this comment!
- Statistical analyses with respect to various quantitative and objective metrics MUST be included. The authors made NO revisions with respect to this comment TOO!
- Only ONE of the suggested relevent and important references is added in the Introduction. No discussion is added even with respect to this solely added reference.
- As stated in my previous review report, there are SUBSTANTIAL typos, language mistakes, and style errors in the manuscript. Even the reference list is a TOTAL MESS! Except for those errors that I have pointed out in the previous round of review, I do not see ANY reduction in such non-technical errors!

Author Response
Reviewer 3#
-The experimental validation is STILL superficial. The authors just added ONE sentence describing data from another totally different experiement as a "comparison", which is not convincing AT ALL!. Only ONE of the suggested relevent and important references is added in the Introduction. No discussion is added even with respect to this solely added reference.
Ans : We have compared with CCEIT developed by Yuxin Wang
The CCEIT developed by Yuxin Wang [29] uses a total impedance of two-phase gas-liquid flow. This research produces real, imaginary, impedance images, and combines real and imaginary parts. Images reconstructed from various impedances are investigated and discussed. The LBP reconstruction method is used to obtain object images from various impedance parts and K-means are adopted to obtain the gray level threshold automatically. The weighted combination method was introduced to combine images by combining images reconstructed from real and imaginary impedances.
Our research obtains impedance, real and imaginary images. All images are obtained from potential data and phases that can be collected from the developed system. LBP reconstruction method from 2 different frequency data can produce the three images. This is the similarity between our research and CCEIT, but we can observe the objects are not only limited for two phases. The advantage of CCEIT compared to our study is that it is non-contact, so it can be used for objects that have high resistivity on the surface.
- Statistical analyses with respect to various quantitative and objective metrics MUST be included. The authors made NO revisions with respect to this comment TOO!
Ans : we have calculated The relative image errors of the images of each part of the
Impedance, real and imaginary images used standard deviation
Z 0,01621
Real 0,02476
Im 0,02123
